# H2B.V demarcates divergent strand-switch regions, some tDNA loci, and genome compartments in *Trypanosoma cruzi* and affects parasite differentiation and host cell invasion

Juliana Nunes Rosón[1,2,3], Marcela de Oliveira Vitarelli[1,2], Héllida Marina Costa-Silva[1,2], Kamille Schmitt Pereira[4,5], David da Silva Pires[1,2], Leticia de Sousa Lopes[1,2], Barbara Cordeiro[1,2], Amelie J. Kraus[6,7], Karin Navarro Tozzi Cruz[1,2], Simone Guedes Calderano[1,2], Stenio Perdigão Fragoso[4,5], T. Nicolai Siegel[6,7], Maria Carolina Elias[1,2], Julia Pinheiro Chagas da Cunha[1,2]*

**1** Laboratory of Cell Cycle, Butantan Institute, São Paulo, Brazil, **2** Center of Toxins, Immune Response and Cell Signaling (CeTICS), Butantan Institute, São Paulo, Brazil, **3** Department of Microbiology, Immunology and Parasitology, Escola Paulista de Medicina–UNIFESP, São Paulo, Brazil, **4** Department of Bioprocesses and Biotechnology, Universidade Federal do Paraná, Curitiba, Brazil, **5** Laboratory of Molecular and Systems Biology of Trypanosomatids, Carlos Chagas Institute, FIOCRUZ, Curitiba, Brazil, **6** Division of Experimental Parasitology, Faculty of Veterinary Medicine, Ludwig-Maximilians-Universität in Munich, Munich, Germany, **7** Biomedical Center, Division of Physiological Chemistry, Faculty of Medicine, Ludwig-Maximilians-Universitäat in Munch, Munich, Germany

* julia.cunha@butantan.gov.br

**Data Availability Statement:** The mass spectrometry proteomics data have been deposited

## Abstract

Histone variants play a crucial role in chromatin structure organization and gene expression. Trypanosomatids have an unusual H2B variant (H2B.V) that is known to dimerize with the variant H2A.Z generating unstable nucleosomes. Previously, we found that H2B.V protein is enriched in tissue-derived trypomastigote (TCT) life forms, a nonreplicative stage of *Trypanosoma cruzi*, suggesting that this variant may contribute to the differences in chromatin structure and global transcription rates observed among parasite life forms. Here, we performed the first genome-wide profiling of histone localization in *T. cruzi* using epimastigotes and TCT life forms, and we found that H2B.V was preferentially located at the edges of divergent transcriptional strand switch regions, which encompass putative transcriptional start regions; at some tDNA loci; and between the conserved and disrupted genome compartments, mainly at trans-sialidase, mucin and MASP genes. Remarkably, the chromatin of TCT forms was depleted of H2B.V-enriched peaks in comparison to epimastigote forms. Interactome assays indicated that H2B.V associated specifically with H2A.Z, bromodomain factor 2, nucleolar proteins and a histone chaperone, among others. Parasites expressing reduced H2B.V levels were associated with higher rates of parasite differentiation and mammalian cell infectivity. Taken together, H2B.V demarcates critical genomic regions and associates with regulatory chromatin proteins, suggesting a scenario wherein local chromatin structures associated with parasite differentiation and invasion are regulated during the parasite life cycle.

to the ProteomeXchange Consortium via the PRIDE partner repository with the dataset identifier PXD026339. http://www.ebi.ac.uk/pride/archive/projects/PXD026339 The ChIP-seq data are deposited at accession SRA data: PRJNA733819 https://www.ncbi.nlm.nih.gov/bioproject/PRJNA733819.

**Funding:** This work was supported by JNR - CAPES, #2019/16033-1, São Paulo Research Foundation (FAPESP); MOV - #2019/04483-2, São Paulo Research Foundation (FAPESP);HMCS - #2019/19834-5, São Paulo Research Foundation (FAPESP); BC - Fundação Butantan; MCE - #2013/07467-1, #2016/50050-2, São Paulo Research Foundation (FAPESP) and Conselho Conselho Nacional de Desenvolvimento Científico e Tecnológico (CNPQ); JPCC #2017/06104-3, #2018/15553-9, #2013/07467-1 from the Sao Paulo Research Foundation (FAPESP). The funders had no role in study design, data collection and analysis, decision to publish, or preparation of the manuscript.

**Competing interests:** The authors have declared that no competing interests exist.

## Author summary

Trypanosomatids have to adapt to different environmental conditions, changing their morphology, gene expression and metabolism. These organisms have many unique features in terms of gene expression regulation. The genomic organization includes polycistronic regions with the absence of well-defined transcription start sites. In *T. brucei*, histone variants mark the start and ending sites of transcription; however, little is known about whether these proteins change their genome location, expression levels and interactors along life forms and what the impact is of these changes on parasite differentiation and infection. In *T. cruzi*, the causative agent of Chagas disease, we previously found that the histone variant of H2B is enriched in TCT forms, a nonreplicative and infective form, suggesting that this variant may contribute to the differences in chromatin structure and global transcription rates observed among these life forms. Here, we aimed to go one step further and performed the first histone ChIP-seq analysis in *T. cruzi*, in which we found that H2B.V was enriched at divergent strand switch regions, some tDNA loci and other critical genomic regions associated with *T. cruzi* genome compartments. We found that H2B.V interacts with a bromodomain factor, suggesting an intricate network involving chromatin acetylation around H2B.V enriched sites. Moreover, parasites expressing reduced H2B.V levels were associated with higher rates of differentiation and mammalian cell infectivity.

## Introduction

Chromatin is formed from the interactions among DNA, RNAs and proteins. Histones are responsible for establishing the folded chromatin structure, presented as nucleosomes, where each one contains two dimers of H2A-H2B and one tetramer of H3-H4. Changes in chromatin conformation caused by histone posttranslational modifications (PTMs) and variant histone deposition can affect interactions in chromatin structure and gene expression [1,2]. Histone variants, aside from canonical histones, differ in primary structure and may be located in specific genomic regions and tissues. For example, CENP-A is a histone variant of histone H3 that is specifically located at centromeres [3]. H2A.Z is present in diverse organisms, destabilizes nucleosome structure and is implicated in transcription activation [4,5]. Few H2B variants have been identified in eukaryotes, including TH2B, which is involved in the cell cycle of germinative mouse male cells [6]; a H2B.Z involved in gene expression in apicomplexans [7], and a H2B.V in trypanosomatids [8], which will be further explored below.

Trypanosomatids are phylogenetically located at one of the deepest branches in eukaryote lineages and include protozoan parasites that cause important human diseases. These organisms have unique characteristics that are mainly related to gene expression and genome structure. Protein-coding genes are organized in long polycistronic transcription units (PTUs) resulting in scenario where posttranscriptional mechanisms play a major role in gene expression [9]. Transcription initiates preferentially in interpolycistronic regions located at divergent strand switch regions (dSSRs) and terminates at interpolycistronic regions located at convergent strand switch regions (cSSRs) [10–12].

Strikingly, trypanosomatids contain variants for all core histones, including a H2B.V [13–15]. In *T. brucei*, histone variants have been shown to play a major role in defining transcription start regions (TSRs) and transcription termination regions (TTRs). H2A.Z and H2B.V dimerize and are frequently located at TSRs together with BDF3, H4K10ac and H3K4me3, forming an active chromatin region for RNA Pol II transcription [13]. In this organism, the

TSRs located at dSSRs are GT-rich and are more prone to promoting H2B.V-H2A. Z deposition and transcription initiation [16]. H3.V and H4.V are frequently located at TTRs together with enrichment of J bases [13,17].

The genome of *Trypanosoma cruzi*, the etiological agent of Chagas disease, was first sequenced by second-generation sequencing strategies in 2005 [18] and then assembled into 41 *in silico* chromosomes [19]. Compared to the genomes of other trypanosomatids, such as those of *T. brucei* and *Leishmania* spp., the *T. cruzi* genome contains more repetitive sequences, mainly composed of multigenic family members (such as trans-sialidases, mucins, MASP, GP63, RHS and DGF-1), which harbor almost 30% of their genome [20], making genome assembly challenging. To overcome this problem, the genomes of two *T. cruzi* strains (TCC and Dm28c) were recently sequenced into long reads generated by third-generation sequencing technologies [21], allowing an accurate estimation of gene abundance, length, and distribution of repetitive sequences. One striking observation was that the *T. cruzi* genome is composed of two compartments that differ in gene composition and have opposite GC contents. The disrupted genome compartment contains the majority of non syntenic genes of trans-sialidases, mucins, and MASPs that are important virulence factors [22], while the conserved compartment contains the syntenic genes of conserved and hypothetical conserved genes.

Similar to other trypanosomatids, *T. cruzi* is able to adapt and survive in different environmental conditions, requiring fast alterations in morphology, metabolism and gene expression [23–25]. Changes in nuclear and chromatin structure together with a global change in transcription rate follow the differentiation of replicative and noninfective forms (epimastigote and amastigote) to nonreplicative and infective forms (tissue culture-derived trypomastigote and metacyclic trypomastigote (TCT and MT, respectively) [26,27]. Furthermore, the nucleosome landscape of epimastigotes and TCTs differs mainly at dSSRs [28]. In addition to these alterations, parasite histones are differentially modified by methylation, acetylation and phosphorylation during the cell cycle and differentiation [29–35].

Previously, we found that H2B.V is differentially abundant in chromatin extracts from epimastigotes and TCT *T. cruzi* forms [36], suggesting that this variant might contribute to the differences in chromatin structure and global transcription rates observed among these life forms [26]. As *T. brucei* H2B.V is deposited at TSRs [13], we hypothesized that *T. cruzi* H2B.V might play a critical role in parasite life forms by modulating chromatin structure and gene expression. Thus, here we aimed to explore this hypothesis by evaluating H2B.V interactors, and by searching for H2B.V genomic location and parasite phenotypic changes along the cell and life cycle using CRISPR-tagged and knockout parasites. H2B.V was shown to be enriched at dSSRs and other critical genomic regions. Moreover, parasites expressing reduced H2B.V levels were associated with higher rates of differentiation and mammalian cell infectivity.

## Results

### H2B.V demarcates *T. cruzi* divergent switch strand regions, some tDNA loci and genome compartments

H2B.V is present as a single-copy gene at chromosome 27 in the *T. cruzi* CL Brener genome assembly. Thus, to evaluate the genomic location of H2B.V, we generated parasites expressing tagged H2B.V at the C-terminus with 3 x myc peptides by CRISPR-Cas9 methodology (S1A Fig). The genomic edition of the H2B.V locus was confirmed by western blotting and immunofluorescence assay (S1C and S1G Fig). Allele-specific PCRs indicate that both alleles were edited (S1B Fig). The genome edition neither affects parasite growth nor the metacyclogenesis processes (S1D and S1E Fig). In addition, H2B.V transcript abundance in epimastigotes and

TCTs do not differ from wild-type cells or parasites expressing Cas9 (S1F Fig). Together, these data indicated that the genomic edition at H2B.V locus confers no significant phenotypic changes.

To identify the genomic regions associated with nucleosomes harboring H2B.V, the chromatin of myc-tagged epimastigotes and TCTs was digested with micrococcal nuclease (MNase) and sonicated followed by chromatin immunoprecipitation (ChIP) using anti-myc and anti-histone H3 antibodies (as a control for the nucleosome distribution along genome). Furthermore, the input for all ChIP samples and an additional control sample of an untagged epimastigote cell line (immunoprecipitated with an anti-myc antibody) were also analyzed. To improve the data analysis and biological conclusions, the reads were mapped against two *T. cruzi* genome assemblies (the Esmeraldo-like haplotype of the CL Brener strain and against the TCC strain), and the results obtained for each were compared throughout this study. The first corresponds to the strain used for ChIP experiments, the genome of which was sequenced by second-generation sequencing strategies [18], and the second harbors 99.7% identity to the CL Brener genome and was recently sequenced by PacBio into long reads [21]. By mapping the reads to these two assemblies, we could evaluate whether any bias related to genome sequencing and assembly quality could interfere with the interpretation of our results. Approximately 2 million reads per replicate per ChIP-seq experiment were sequenced, generating approximately 85% and 90% overall alignment (S2A Fig) in the CL Brener and TCC assembly, respectively.

Visual inspection of H2B.V enrichment (Fig 1A) indicated that this variant was deposited along the edges of dSSRs, including, in some cases, the first few coding DNA sequences (CDSs) from the polycistronic regions (Figs 1A and S3A). In accordance, the summary plot of the normalized ratio of H2B.V (ChIP/input) enrichment clearly indicated that H2B.V was preferentially enriched in the upstream region of polycistrons (Fig 1B) in both the CL Brener

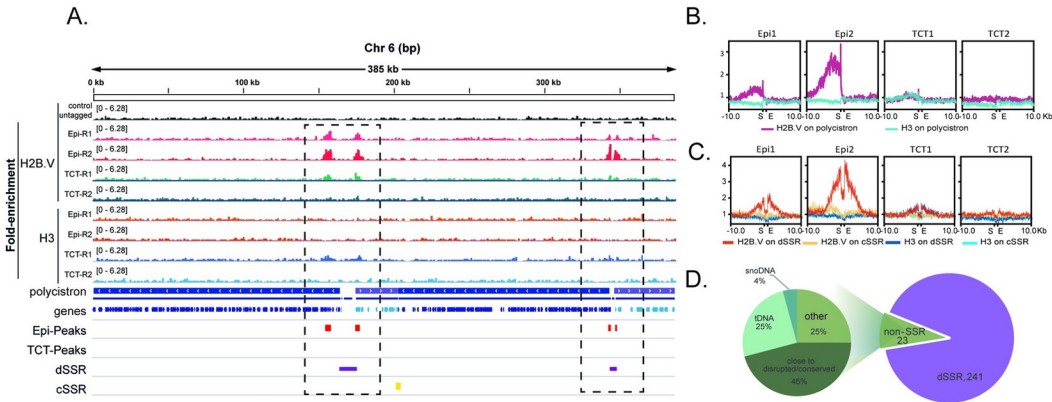

**Fig 1. H2B.V is enriched at dSSRs of epimastigote forms.** A. IGV snapshot of whole chr 6S of the CL Brener Esmeraldo-like haplotype using the wig files from the ChIP H2B.V/input and ChIP H3/input ratios obtained by COVERnant v.0.3.2. Note that H2B.V, but not H3, is enriched at dSSRs. Blue bar arrows indicate the transcription direction in each polycistron. Genes in the same polycistron are stained with the same color. dSSRs, purple bars; cSSRs, yellow bars. Red bars represent H2B.V peaks (fold equal to 4 over the input, Poisson p-value over input required = 1.00e-04) obtained by HOMER in the epimastigote form. Rectangles with interrupted lines highlight dSSRs. Summary plot obtained by deepTools for H2B.V and H3 (ChIP/input) enrichment at polycistrons (B), dSSRs and cSSRs (C). Ten kilobases upstream and downstream of the abovementioned regions were also analyzed. All genomic features were automatically scaled as indicated (Start-End). Polycistron start and end is related, respectively, to the first ATG (from the first CDS) and the last stop codon (from the last CDS). dSSR and cSSR start refer to the first coordinate downstream, respectively, to the start and end of the left polycistron. dSSR and cSSR end refer to the first coordinate upstream, respectively, to the start and end of the right polycistron. D. Percentual distribution of H2B.V peaks (Fold 4) on all 41 chromosomes of the CL Brener Esmeraldo-like haplotype.

(Fig 1B) and TCC assembly (S3B Fig). Through H2B.V peak enrichment data, we observed that H2B.V was deposited over large regions of approximately 5 kb in width (Figs 1C and S2C). dSSRs, but not cSSRs, were demarcated by a peak at each edge, and no specific histone H3 enrichment was found at these regions (Fig 1C).

We used a peak calling algorithm to faithfully identify and compare peaks among life forms and genome features (S2B and S2D Fig and S2 and S3 Tables). Peaks were obtained for each replicate separately, but only the overlapping peak set was used for further analysis (S2B and S2D Fig). From the 265 peaks found in epimastigotes (fold change > = 4, Poisson p-value over input required = 1.00e-04), 91.3% were located at dSSR, and only 8.7% were found at non-SSR in the CL Brener assembly (Fig 1D and S2 Table). A higher enrichment of H2B.V at dSSR (76%) over non-SSR (24%) was also found in the TCC assembly (S3C Fig).

Although the great majority of H2B.V peaks were located within a dSSR, not all dSSRs contained H2B.V enrichment (S4A Fig). Specifically, 38% and 68% of dSSRs did not have any H2B.V peak (considering a fold change (ChIP/input) of 2) in the CL Brener and TCC assembly, respectively. However, these dSSRs usually flank monocistrons (67% and 60% in the CL Brener and TCC assembly, respectively); small (fewer than 4 CDSs) polycistrons (7.5% and 12% in CL Brener at TCC, respectively); snoDNA, tDNA (21% in the TCC assembly) (S5A Fig) or rDNA (S5B Fig). In the CL Brener assembly, we additionally noticed that 20% of the dSSRs that did not harbor the H2B.V peak were located close to a gap assembly region. In short, only 5% and 6% of dSSRs located between two protein coding polycistronic regions (with more than 4 CDSs) lacked H2B.V enrichment.

Among the H2B.V peaks located at non-SSRs, we found that they were closely associated with tDNA genes (25% and 5% in CL Brener and TCC, respectively), snoRNA genes (4% in CL Brener and 6% in TCC), and, more strikingly, between the disrupted and conserved genome compartments (46% -11 out of 23, in the CL Brener assembly) (Figs 1D, 2A, S3C and S4A). In the TCC assembly, 45% of the non-SSR H2B.V peaks were between disrupted and conserved compartments (S3C and S4B Figs).

To further explore this phenomenon, we evaluated the enrichment of H2B.V among multigenic family members composing the disrupted genome compartments (trans-sialidases, MASP, and mucins) and those that could be part of both conserved and disrupted compartments (GP63, RHS, and DGF-1). Heatmaps of H2B.V ChIP data indicated that members of disrupted families were more enriched in H2B.V (Fig 2B). Hierarchical clustering of all multigenic family members showed that mucins and MASP were preferentially enriched on H2B.V (14 and 15%, respectively, considering each gene class individually) (S4B and S4C Fig). Interestingly, evaluation of these two gene families in the most recent genome assembly with long reads [21] suggested that they are preferentially located at the first CDSs of a given PTU, which in turn would partially explain the enrichment of H2B.V in these gene families.

A considerable number of non-SSR H2B.V peaks were located close to tDNA genes (Figs 2A and S5A). Thus, we evaluated the enrichment of H2B.V in tDNA loci by deeptools analysis (Fig 2C). H2B.V was enriched either at upstream or downstream regions of tDNA genes, with no clear association with tDNA anticodon type or adjacent polycistron transcription direction. tDNA genes were present alone or in clusters of 2–10 genes totaling 21 loci in the CL Brener Esmeraldo-like haplotype. Most of them were located within a protein-coding polycistron, suggesting that tDNA loci might interfere with RNA Pol II transcription elongation. Of the 10 loci of tDNA located at non-SSRs, seven were associated with a H2B.V peak (fold >2); from the six tDNA loci located at dSSR, all were associated with an H2B.V peak, while 2 out of 4 located at cSSR had an H2B.V peak.

Taken together, H2B.V was enriched at dSSRs flanking protein-coding polycistrons, at some tDNA loci, and at regions between the disrupted and conserved genome compartments.

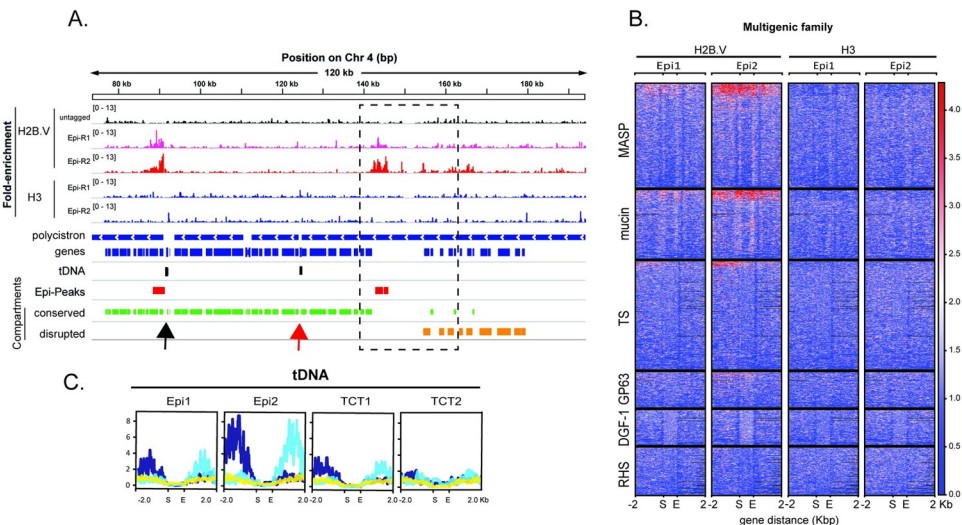

**Fig 2. H2B.V enrichment is also found at the border of conserved and disrupted genome compartments and around tDNA loci.** A. IGV snapshot of a 120-kb region from Chr 4S highlighting H2B.V enrichment around tDNA loci (black arrow) and between conserved and disrupted genome compartments (rectangle with interrupted lines). Wig files from the ChIP H2B.V/input and ChIP H3/input were obtained by COVERnant. Blue bar arrows indicate the transcription direction in each polycistron. Genes in the same polycistron are stained with the same color. Black and red bars represent respectively, tDNA loci and H2B.V -peaks (fold equal to 4 over the input, Poisson p-value over input required = 1.00e-04) obtained by HOMER in epimastigote form. The conserved and disrupted compartments are represented by green and orange bars, respectively. B. Heatmaps of multigenic family members showing ratios of ChIP H2B.V/input and ChIP H3/input in two epimastigote replicates. Note an enrichment (red) of H2B.V at MASP, mucin and TS genes. C. Summary plot obtained by deepTools of the normalized (ChIP/input) H2B.V reads at tDNA loci and the adjacent 2 kb upstream and downstream regions. The tDNA locus was automatically scaled as indicated (first (S) and last (E) nucleotides of each locus). Normalized data were classified into 3 hierarchical clusters (dark blue– 13 genes, light blue– 11 genes and yellow -31 genes) based on the H2B.V enrichment pattern.

In contrast, dSSRs mainly associated with monocistrons and small polycistrons did not have H2B.V enrichment.

## H2B.V Chip-seq enrichment differs greatly among epimastigote and TCT forms

Previously, H2B.V was found to be enriched in TCT chromatin when similar masses (in micrograms) of chromatin proteins released upon DNase treatment, from both life forms were compared [36]. We confirmed that H2B.V was enriched in TCT whole cell extracts (WCE) when equal numbers of epimastigotes and TCT parasites (in biological quadruplicates) were compared, while no important difference was found for histone H3 (S6A Fig).

Intriguingly, fewer sites of H2B.V enrichment were found along the TCT genome when compared with epimastigotes (Figs 1A, S2B and S3A). While 265 H2B.V-enriched peaks were found at epimastigotes, and only 6 peaks were found at TCTs (fold change over the input > = 4) in the CL Brener assembly. Using the same parameters, fewer and smaller (in width, in bp) peaks were found in TCT life forms than in epimastigote forms in the TCC assembly (S2B and S2C Fig). This finding agrees with lower H2B.V staining in myc-tagged TCTs nuclei detected by IFA (S1G Fig) but contrasts with previous findings [31].

Thus, we hypothesize that H2B.V may weakly interact with the TCT's chromatin and be released upon experimental procedures like IFA and ChIP. To address this, we performed a chromatin sequential salt extraction assay, which evaluates the relative binding affinities of chromatin-associated proteins [37]. The results indicated that H2B.V (and H3) is released

from chromatin in lower salt concentrations in TCTs than epimastigote forms, suggesting that H2B.V has a lower binding affinity to TCTs chromatin (S6B Fig).

As a whole, these findings suggest that the parasite modulates H2B.V enrichment in life forms, as will be discussed below.

## H2B.V interacts with a bromodomain factor and nucleolar proteins

To gain better insights into the role of H2B.V in chromatin, we evaluated its interaction partners in epimastigotes and TCTs by pulldown assays. Therefore, recombinant H2B.V and canonical H2B (as a bait control) were used to obtain specific H2B.V interactors using the WCE of epimastigotes and TCTs. In addition, four control samples (described in detail in the Materials and Methods section) were also analyzed, which included the analysis of both parasite WCE and the recombinant protein extracts upon affinity resin incubation. From here, all experiments were performed with the *T. cruzi* Y strain for comparison with previous results [36].

Protein eluates were processed for mass spectrometry, and the results were subjected to *in silico* filtering steps (described in Materials and Methods) to obtain a more meaningful protein set of H2B.V interactors (Fig 3A and S4 and S5 Tables). From the final list of 39 interactors, two proteins were present in all pulldown eluates: double RNA binding domain protein 3 (TcCLB.506649.80) and polyadenylate-binding protein (TcCLB.508461.140), which is known to interact selectively and noncovalently with a sequence of adenylyl residues in an RNA molecule and participate in mRNA maturation at the poly(A) tail [38]. Interestingly, H2B.V and H2B interacts more with proteins found at TCTs than at epimastigote extracts. Eight proteins were found to interact with H2B.V in epimastigotes, whereas 26 proteins were found to

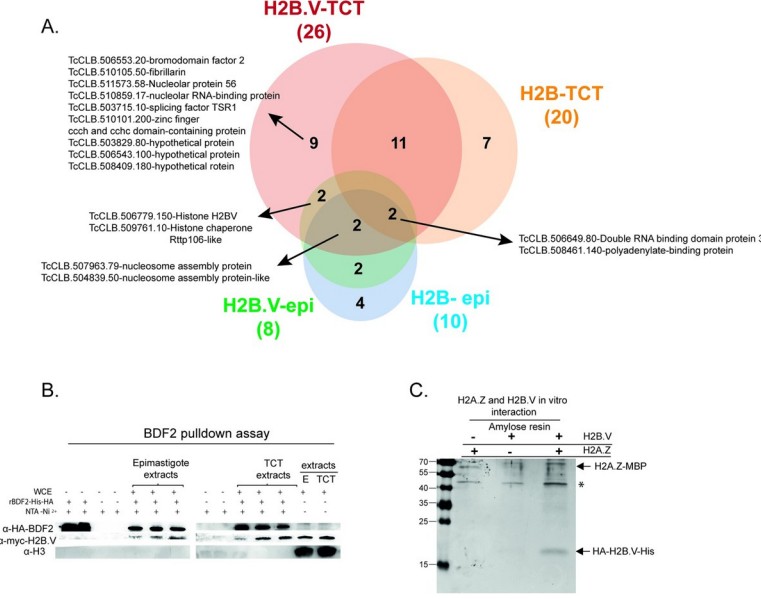

**Fig 3. H2B.V *in vitro* interactors.** A. Venn diagram showing the number of common and specific interactors of H2B. V and H2B using epimastigote and TCT extracts. The total number of eluted proteins in each pulldown is shown between parentheses. Some relevant proteins are highlighted. B. BDF2 pulldown assays. Recombinant BDF2 tagged with HA and 6xHis was incubated with NTA-Ni resins with or without H2B.V-myc-tagged parasite extracts (epimastigotes or TCTs). Eluates were fractionated by 15% SDS-PAGE, and protein interactors were revealed by WB with anti-HA (against rBDF2), anti-myc (against H2B. V-myc tagged), and anti-H3 (as a chromatin control). C. H2A. Z interacts with H2B.V *in vitro*. H2A.Z-MBP and H2B.V-6xHis recombinants were incubated together or separately with amylose resin, known to specifically interact with MBP. Eluates were fractionated by 15% SDS-PAGE and stained with Coomassie. H2A.Z-MBP is approximately 60 kDa, and H2B.V-6xHis is 16.5 kDa. The * indicates a contaminant of lysed extraction.

interact with H2B.V in TCTs. Six were common to both, including 2 nucleosome assembly proteins (TcCLB.507963.79 and TcCLB.504839.50) and a hypothetical protein (TcCLB.509761.10) that was recently annotated as histone chaperone Rttp106-like, known in *S. cerevisiae* for being associated with DNA replication and heterochromatin silencing [39]. Interestingly, the latter was not found in canonical H2B pulldowns, suggesting that it might be associated with H2B. V nucleosome assembly.

Among H2B.V and H2B interactors in TCT extracts, we found a high mobility group protein TDP1 (TcCLB.507951.114), which recognizes and links noncanonical DNA structures, such as cruciform DNA and circular mini-DNA, in addition to its involvement in DNA folding [40,41]; a lupus La protein homolog (TcCLB.511367.60), which is an mRNA ligand and participates in mRNA maturation and translation, and stabilization of histone mRNAs during the S phase in humans [42,43], in addition to binding the 3′ poly(U)-rich in nascent RNA polymerase III transcripts participating in its correct folding and maturation in *T. brucei* [44]; a retrotransposon hot spot (RHS) protein (TcCLB.506113.60); and tousled-like kinase II (TcCLB.510597.9). The latter has enzymatic activity and is involved in several biological processes, such as DNA replication and S-phase progression, repair and H3S10 phosphorylation [45,46].

One of the most interesting findings was the presence of 3 nucleolar proteins specifically in H2B.V-TCT pulldowns: nucleolar protein 56 (TcCLB.511573.58), fibrillarin (TcCLB.510105.50), nucleolar RNA-binding protein (TcCLB.511573.58). In addition, two other hypothetical proteins (TcCLB.508409.180 and TcCLB.503829.80) were also detected, which were previously annotated as proteins of the nuclear pore and nuclear envelope, respectively [47,48]. Among H2B.V-specific interactors, we identified protein bromodomain factor 2 (BDF2) (TcCLB.506553.20), an epigenetic factor that recognizes histone acetylation [49,50]. In *T. cruzi*, BDF2 has already been shown to interact with histone H4 acetylated residues (K10 and K14), showing increased levels after exposure to UV in epimastigote forms and likely being involved in the response to DNA damage [51].

The H2B.V and BDF2 interaction was validated by reverse pulldown assays using recombinant BDF2-6xHis-HA and WCE of H2B.V-myc-tagged parasites (epimastigotes and TCTs) (Fig 3B). In contrast to what was observed in H2B.V pulldowns, here the BDF2 interacted with H2B.V in both life forms. In addition, the BDF2 interaction with H2B.V seemed to be specific and not a spurious histone interaction, as histone H3 was not eluted from the BDF2 pulldown. Nevertheless, BDF2 interaction with other core histones should be evaluated to confirm this specific interaction.

It is worth mentioning that besides H2B.V and H2B, we only detected H4 peptides in these pulldown assays. Here, the proteomics protocol was not optimized for histone identification, which is often overdigested by trypsin due to high amounts of lysine and arginine residues. In *T. brucei*, H2B.V dimerizes with the variant H2A.Z [52] and is preferentially detected at dSSRs. To verify whether *T. cruzi* H2B.V also dimerized with H2A.Z, we cloned and expressed the recombinant H2A.Z (TcCLB.511323.40) fused to a maltose binding protein (MBP) and incubated it with the 6xHis-tagged H2B.V recombinant (Fig 3C). Upon elution with maltose, both H2A.Z-MBP (~60 kDa) and H2B.V-6xHis (~16.5 kDa) were eluted, indicating that H2A. Z-H2B.V also interacted in *T. cruzi*. This *in vitro* interaction was further confirmed by Western blot analysis using polyclonal antibodies against H2A.Z (S7 Fig).

## H2B.V is essential but H2B.V heterozygous knockout (H2B.V-HtzKO) epimastigotes do not display alterations in growth, replication or global transcription levels

To further understand the biological role of histone H2B.V in *T. cruzi*, we generated H2B.V knockout parasites by homologous recombination. Therefore, constructs containing

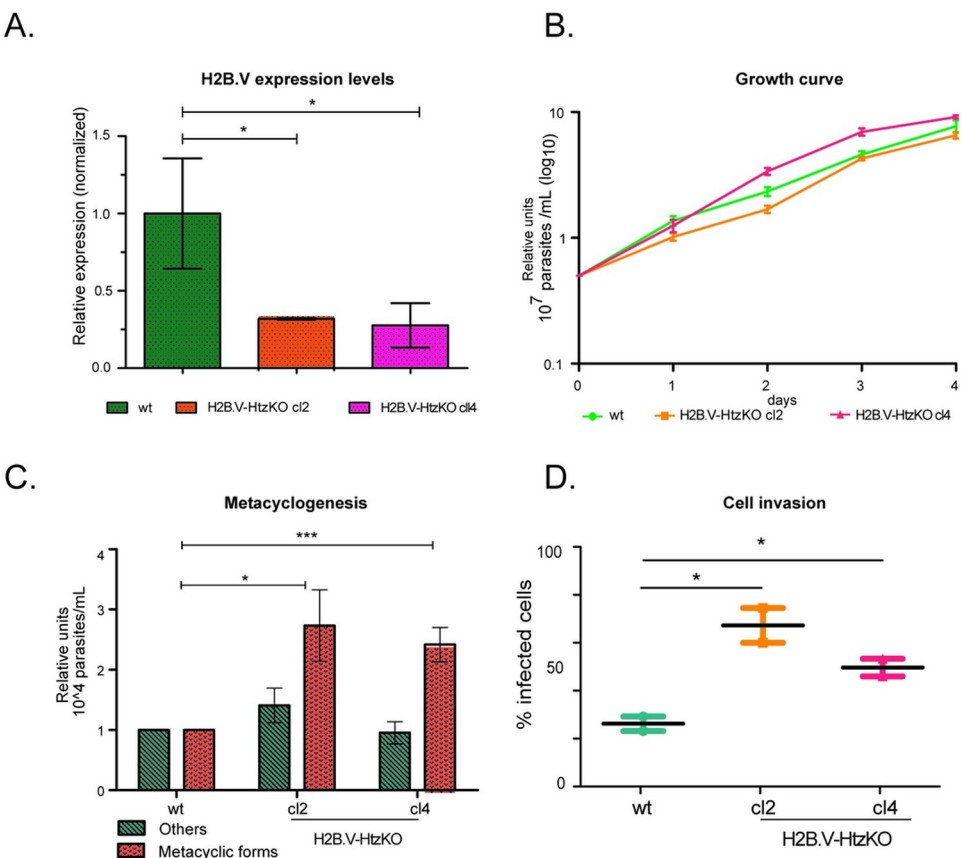

**Fig 4. Phenotypic evaluation of H2B.V-HtzKO parasites.** A. H2B.V relative expression levels were evaluated by qPCR in wild-type and H2B.V-HtzKO (cl 2 and 4) parasites. H2B.V transcript levels were assessed relative to GAPDH levels (ΔCtH2BV−ΔCtGAPDH). One-way ANOVA: * p-value < 0.05. B. Mean growth curves (in log10) for wild-type and H2B.V-HtzKO (cl 2 and 4) parasites. C. Metacyclogenesis assay. The number of metacyclic trypomastigotes, epimastigotes and intermediate forms (herein, classified as other) were counted in 96-h supernatant upon nutritional stress as described in the Materials and Methods. Values were normalized to their respective numbers in wild-type cells. Experiments were performed in biological triplicates. Unpaired T-test: *p < 0.05 and ***p < 0.01 with Bonferroni correction. D. H2B.V-HtzKO parasites are more infective. LLCMK2 cells were infected with TCTs (1:40; cells:parasite). Slides were fixed, and nuclei were labeled with DAPI. The number of infected cells was counted relative to the total cells after 1 h of infection by fluorescence microscopy. Unpaired t-test with Welch's correction *** p < 0.0001.

neomycin resistance genes flanked by sequences from the 5' and 3' UTRs of the H2B.V gene (NEO cassette) were generated (S8A Fig) and used to obtain parasites that had lost one H2B.V allele and gained resistance to neomycin. The correct insertion of the neomycin cassette into the H2B.V locus was confirmed by PCR using specific primers (S8B Fig). Many attempts to delete the second H2B.V allele and generate double knockout (or homozygous knockout) parasites using a similar strategy described above (but with a cassette containing the hygromycin resistance gene–described in Materials and Methods) were performed without success. Thus, these results suggested that the H2B.V gene must be essential for *T. cruzi*, similar to *T. brucei* [52]. Thus, all phenotypic analyses described below were performed using two clones (cl 2 and 4) isolated from the transfected parasite population of HtzKO.

First, we confirmed that H2B.V transcription levels were decreased in HtzKO parasites by qPCR (Fig 4A) and by quantitative proteomics assays (iBAQ values) using basic WCE (S8C Fig). H2B.V-HtzKO cl2 and cl4 had, on average, 4 times less H2B.V protein than wild-type

cells (unpaired t-test, p-value < 0.05). In accordance, the levels of histone H4 and all other histones did not change significantly in H2B.V-HtzKO mice. Then, we evaluated whether the lower levels of H2B.V in H2B.V-HtzKO parasites interfered with growth and cell cycle progression. No statistically significant differences were found among H2B.V-HtzKO clones and wild-type parasites when growth and cell cycle phases were evaluated (Figs 4B and S9A–S9C).

As H2B.V was enriched at putative TSRs, we asked whether H2B. V-HtzKO parasites would show a decrease in global transcription by evaluating EU incorporation. Again, no significant global differences were found among clones and wild-type parasites (S9D Fig).

The reduced H2B.V levels might result in a more stable chromatin structure once nucleosomes containing H2A.Z-H2B.V are more unstable than canonical nucleosomes [13]; thus, we evaluated whether H2B.V-HtzKO parasites would have global changes in chromatin structure by evaluating the amount of active chromatin extracted by the FAIRE approach [53]. H2B. V-HtzKO parasites and wild-type parasites have similar levels of open chromatin on H2B. V-HtzKOs (S9E Fig).

Since we evaluated global effects on transcription and active chromatin levels, we could not rule out that specific genomic regions could be affected in H2B.V-HtzKO parasites. Future transcriptomic and sequencing analyses from the data shown in S9D and S9E Fig could clarify this issue.

## Changes in H2B.V levels can affect metacyclogenesis and infection of mammalian cells

To assess whether the decrease in H2B.V levels in H2B.V-HtzKO parasites affect the differentiation of epimastigotes into metacyclic forms, we performed a metacyclogenesis assay evaluating both the presence of fully differentiated metacyclics and their intermediate forms (based on kinetoplast morphology and position relative to the nucleus as previously described [25]. Labeling of the metacyclic membrane markers GP90 and GP82- was also used to discriminate differentiated parasites [54].

In the stationary growth phase, H2B.V-HtzKOs (cl2) had approximately 60% more intermediate parasite forms than wild-type parasites (S9F Fig), suggesting that changes in H2B.V levels might facilitate the differentiation of epimastigotes into metacyclic trypomastigotes. Furthermore, in the 96-h supernatant after the induction of metacyclogenesis, almost 3 times more trypomastigote metacyclics were found in H2B.V-HtzKO when compared to wild-type parasites (Fig 4C). This supernatant was also enriched in non-metacyclic parasites (classified as "others", which included epimastigote and intermediate forms), suggesting that changes in H2B.V levels promoted loss of adhesion to substrates.

We also evaluated whether the ability to infect mammalian cells would be affected in H2B. V-HtzKOs TCTs. Surprisingly, we found that cl2 and cl4 infected approximately twice as many cells as wild type (Fig 4D). Taken together, these data suggested that decreasing H2B.V levels might interfere with parasite differentiation and infection of mammalian cells.

## Discussion

H2B has few variants described in eukaryotes, but trypanosomatids have a H2B.V that has been shown to dimerize with H2A.Z [52] and to be predominantly located at dSSRs [13]. Here, we investigated the genomic location, interaction partners, and phenotypic changes in heterozygous knockout H2B.V parasites. To our knowledge, our work is not just the first histone ChIP-seq analysis performed in *T. cruzi* but also the first to compare, from a genome-wide perspective, the differences in trypanosomatids of two life forms (precisely, epimastigote

and TCT) that share key differences associated with replication and infection capacity, shedding light on epigenetic changes in these organisms, as discussed above.

We found that H2B.V was preferentially located at *T. cruzi* dSSR edges extending ~5 kb and included the first few CDSs at a polycistron. Importantly, H2B.V enrichment was also found at a few non-SSRs, which were preferentially close to tDNAs and, strikingly, between the disrupted and conserved genome compartments, with enrichment mainly in mucin and transsialidase genes. Moreover, the great majority of dSSRs flanking polycistrons containing protein-coding genes showed an enrichment of H2B.V. In contrast, dSSRs flanking monocistrons and short polycistrons (less than 4 CDSs) did not contain a significant enrichment of H2B.V. Future studies should clarify why these dSSRs are not enriched in H2B.V and whether its absence has any transcriptional consequences for the flanked polycistron.

The comparison of ChIP-seq results in the two genome assemblies obtained by the 2nd (CL Brener strain) or 3rd (TCC strain) generation sequencing strategies reinforces the preferential location of H2B.V at dSSRs, tDNAs, and at the border of the disrupted and conserved compartment. However, this comparison also highlighted differences: more dSSRs harbor no H2B. V enrichment in TCC assembly (68% versus 38% in CL Brener). Among the non-SSR peaks, only 5% of them lay on tDNAs in TCC (compared to 25% in CL Brener). The difference could be related to a better mapping and assembly resolution obtained by the 3rd generation sequencing or by the biological differences between strains. In addition, we detected enrichment of H2B.V in mucins and MASP genes. However, analysis at new assemblies' genomes indicates that these genes are located preferentially at the first CDS of a given PTU, thus close to dSSRs. These findings strengthen the necessity of comparing the results in the two genome assemblies to assess data reliability and reproducibility. This comparison seems more critical for biological conclusions made at the multigenic family loci, which were better assembled in the 3rd generation sequencing strategies.

It is well known that global transcription levels are decreased in TCTs in relation to epimastigote forms [26], and similar findings have been found in *T. brucei* [55,56]. Recently, we observed that dSSRs in TCTs have more nucleosomes, which are less dynamic than those from epimastigote forms [28]. Here, we detect that H2B.V is predominantly located at dSSRs of epimastigotes but very few H2B.V enrichment peaks were found on TCTs' genome. Although the H2B.V protein abundance is higher on TCTs, this histone has a lower binding affinity to their chromatin, as observed by sequential salt extraction assays. Nucleosomes harboring H2B. V-H2A.Z are less stable than canonical nucleosomes [13], and their absence could (partially) explain why dSSRs of TCTs were more occupied by nucleosomes than epimastigotes [28]. We speculate that the absence of histone variants in TCT would result in a more stable chromatin structure around dSSRs that, in turn, could hamper RNA Pol II binding, likely explaining the decreased global levels of transcription in this life form [26].

One question that remains to be answered concerns what induces the differential H2B.V enrichment and binding affinity on *T. cruzi* chromatin among life forms. One explanation could be the presence of specific H2B.V/H2A. Z PTMs for each life form. However, we detected no statistically significant difference between H2B.V PTMs in epimastigotes and TCTs, and we found changes in acetylation levels at H2A.Z peptide [41]GKGKGKGKGKR[51] between epimastigotes and TCTs [32]. Another possibility concerns the expression and half-life of H2B.V and the protein complexes involved in its deposition. The deposition of histone variants in chromatin is usually independent of DNA replication [57], but how it is achieved is still not fully understood. In humans, the mammalian homolog SWC2 (YL1) is related to H2A.Z deposition [58]. In *T. brucei*, the GT-rich promoters and H4 acetylation target H2A.Z deposition [16,59]; however, no protein complex was associated with this function. In our pull-down assays, we detected one histone chaperone (TcCLB.509761.10-histone chaperone

Rttp106-like) that interacted specifically with H2B.V, providing a promising candidate for H2B.V deposition. It remains unclear whether this chaperone indeed has a role and whether its levels are differentially expressed throughout the parasite lifecycle affecting H2B.V deposition.

The interactome data highlighted the presence of pore and nuclear envelope proteins (TcCLB.503829.80 and TcCLB.508409.180), suggesting a preferential location of H2B.V in the nuclear periphery. Furthermore, H2B.V interacted with important nucleolar proteins, which might suggest that H2B.V deposition could also influence RNA pol I activity in addition to RNA pol II. Here, we observed an accumulation of H2B.V along rDNA genes (S5B Fig); however, rDNA assessment is always challenging due to its very repetitive nature. In the TCC genome assembly, for example, 249 rDNA genes were distributed over 11 contigs.

In *T. cruzi*, BDF2 was located in the nucleus at all life cycle stages and was shown to interact with acetylated histone H4 K10 and K14 residues [51]. Here, we found that BDF2 interacted with H2B.V, suggesting a chromatin environment in which acetylated lysines were present. Bromodomain proteins have an important role in chromatin function, since their bromine domain participates in biological processes such as DNA replication, transcription, DNA repair and silencing [60–62]. In *T. brucei*, the dSSR is enriched on BDF3 [13], possibly interacting with the many acetylated histones found at dSSRs [59]. Histone acetylation, histone variant deposition and mRNA levels are intrinsically connected in *T. brucei*. Loss of TSS-H4 acetylation reduces H2A.Z deposition in TSS to shift the RNA initiation site, while H2A.Z acetylation is directly associated with increased global mRNA levels [59]. It remains to be investigated whether BDF2 is important to integrate this scenario in *T. cruzi*. Nevertheless, we have previously found multiple acetylations at H2B.V and H2A.Z in *T. cruzi*, the acetylation levels of which change during metacyclogenesis [32]. It is worth noting that in *T. brucei*, BDF2 has been shown to be associated with H2A.Z N-terminal hyperacetylation [63].

The peculiar genome location of H2B.V together with its interaction partners suggested that H2B.V might be associated with important regulatory functions in *T. cruzi*. Heterozygous knockouts for H2B.V showed no important effects on proliferation, the cell cycle or global transcription rates in epimastigote forms, suggesting that i. these processes are independent of H2B.V regulation; ii. other undescribed protein might fulfill the H2B.V function; or iii. the reduced level of H2B.V in HtzKO clones was insufficient to induce critical changes in the abovementioned processes. We believe that the latter hypothesis is more feasible, as we failed to obtain a double knockout parasite, which suggests that H2B.V is essential for cell viability. Moreover, we observed critical effects on metacyclogenesis and mammalian cell infection in H2BV-HtzKO parasites. These results raise the question of how changes in H2B.V abundance would interfere with these phenomena considering the trypanosome's posttranscriptional regulation scenario [9]. In trypanosomes, the 3D nuclear structure and compartmentalization play a crucial role in the spatial organization of splicing and transcription [64]; moreover, parasites that have lost H4.V and H3.V show profound changes in nuclear architecture [65]. In accordance, we found H2B.V enriched at the border of conserved and disrupted genome compartments. The latter is composed of important virulence factors (trans-sialidase, MASP, and mucin genes) that are mainly expressed in metacyclics forms [23]. Thus, it is tempting to proposed that H2B.V may mark the transition between these two compartments influencing the *T. cruzi* nuclear architecture and gene expression of these virulence factors.

Moonlight proteins are multi-tasking proteins that perform various functions [66]. Recently, histone H3-H4 tetramer was shown to have cupper reductase activity, expanding histone function beyond chromatin compaction [67]. Thus, *T. cruzi* H2B.V may also harbor moonlight functions affecting cell invasion and differentiation. H2B.V-HtzKO parasites easily lost adhesion to substrates and differentiated almost 3 times more into metacyclics than wild-

type parasites. Loss of adhesion to a substrate is critical for epimastigote-to-metacyclic differentiation [68]. However, we could not determine the direct connection between H2B.V and the differentiation/infection capacity, but changes in histone levels and infectivity are not unprecedented. Leishmania cell lines overexpressing histone H1 have lower infectivity both *in vitro* and in vivo [69–71]. More recently, Leishmania histone H3 was shown to interact with human nuclear histones as part of their nucleosomes [72], which reinforces evidence showing that parasite infection may change the host epigenome [73]. Intriguingly, histones are identified on the secretome of *T. cruzi* [74] and could interfere with infection. Here we found that H2B.V has a lower binding affinity to TCT's chromatin. However, it remains to be investigated whether H2B.V plays other functions than compaction in TCT's nuclei. and whether it has any other role in parasite infection.

Our data indicate 23 putative new TSRs at non-SSRs (CL Brener Esmeraldo-like haplotype), in which at least 7 were associated with the presence of a tDNA gene inside a polycistron of CDSs. tRNAs are transcribed by RNA Polymerase III and may interfere with RNA Polymerase II transcription/elongation when located near a CDS [13,75–77]. This interference may have special consequences considering the polycistronic transcription scenario in trypanosomatids. In *T. brucei*, an enrichment of H2A.Z was shown downstream of tDNA loci, likely creating a new TSR [13]. Here, we found a more complex scenario: for 3 tDNA loci located at non-SSRs, no H2B.V enrichment was found, suggesting that the absence of a permissive chromatin structure for transcription initiation may negatively interfere with the transcription of the downstream CDSs. If this is correct, it will indicate an interesting alternative to regulate transcription inside the polycistron.As transcription initiation by RNA pol II in trypanosomatids does not occur on well-defined focused promoters [16], we envisage that these parasites may take advantage of alternative ways to regulate gene expression throughout their life cycle stage by strategically combining the deposition of variant histones such as H2B.V and the location of tDNA genes and nucleosomes in their genome. In addition, the finding that BDF2 interacts with H2B.V may indicate the existence of a complex scenario of histone PTMs at dSSRs that should be explored in the future to highlight further epigenetic modulators, which could be additionally modulated in life forms.

## Materials and methods

### Parasites, metacyclogenesis, growth curves and mammalian cell invasion

*T. cruzi* epimastigotes (Y and CL Brener strains) were cultured at 28˚C in LIT medium supplemented with 10% fetal bovine serum (FBS—Vitrocell), glucose 0.4%, hemin 0.1 μM and penicillin-G 59 mg/L as previously described (Camargo, 1964). H2B.V-HtzKO (cl 2 and 4- Y strain), H2B.V-myc tagged (CL Brener strain) epimastigote forms were maintained in the abovementioned medium supplemented with 500 μg/mL G418 (Gibco) and 500 μg/mL puromycin. For growth curves, epimastigotes were diluted to $5 \times 10^6$ cells/mL and monitored for four days. Cell density was counted in the Z2 Coulter Particle Count and Size Analyzer (Beckman Coulter) using a 100-μm filter. TCT forms (Y and CL Brener strains) were obtained from the supernatant of infected LLCMK2 cells after 1 week of infection (1:40 –cell:parasites) according to [78]. Cells were maintained in DMEM (Gibco) containing $NaHCO_3$ 3.7 g/L, penicillin G 0.059 g/L, streptomycin 0.133 g/L and FBS 10% at 37˚C with 5% CO2. For mammalian cell invasion assays, after 1 h of infection (1:40, cell:parasites), cells were fixed with 4% paraformaldehyde, stained with DAPI and visualized by Olympus IX81 fluorescence microscopy. For ChIP-seq experiments, TCTs samples with a maximal of 5% of amastigotes were used. For Y strain (Wt and H2B.V-HtzKOs) the metacyclogenesis protocol described by [79] was followed by evaluating the TAU 3AAG culture supernatant upon 96–144 h of nutritional

stress. For CL Brener (H2B.V-myc tagged), a culture of ~5 x10^6 epimastigote/mL was maintained undisturbed for 8 days in RPMI medium without FBS, following the protocol described by [80]. Intermediate forms were classified based on kinetoplast and nucleus position, as shown in [25]. Antibodies against GP90 and GP82 were used to facilitate visualization of metacyclic parasites. Parasites were counted in technical triplicates from each biological triplicate.

## Generation of H2B.V-HtzKO parasites

All primers sequences are described in S1 Table. The regions upstream and downstream of the H2B.V gene (TcCLB.506779.150) were cloned from the genomic DNA of the *T. cruzi* Y strain into the recombinant plasmids pTc2KO-neo and pTc2KO-hyg [81]. The upstream region of the H2B.V gene (434 bp) was amplified by PCR using the primers H2B.V_KpnI (forward) and H2B.V_SalI (reverse) (fragment 5'flank_H2B.V). Simultaneously, the 507-bp fragment from the intergenic region downstream of the H2B.V gene was amplified by the primers H2B. V_BamHI (forward) and H2B.V_XbaI (reverse) (fragment 3'flank_H2B.V). The recombinant plasmids pTc2KO-H2B.V-neo and pTc2KO-H2B.V-hyg were purified, and the "NEO cassette" (2,671 bp) and "HYG cassette" (2.971 bp) were amplified by PCR using the primers H2B. V_KpnI (forward) and H2B.V_XbaI (reverse) (S8A and S8B Fig left). Parasite transfection and selection were performed by homologous recombination of the NEO cassette, and parasites resistant to G418 were confirmed by PCR using the primers NEO_F and NEO_R; H2B_EXT_-FOR and H2B_EXT_REV, NEO Out R and NEO Out F (S8B Fig right). The neomycin-resistant parasites were electroporated with the "HYG cassette" (5'FLANK-HYG-3'FLANK) to generate the H2B.V homozygous knockout, which would be resistant to both G418 and hygromycin B (at 500 µg/mL). Clones were obtained by serial dilution by FACS sorting.

## Generation of H2B.V-myc parasites

All primer sequences are described in S1 Table. The CRISPR-Cas9 protocol was based on [82]. Donor DNA and sgRNA were amplified by PCR using high-fidelity Phusion DNA Polymerases (Thermo Fisher Scientific) considering the H2B.V (TcCLB.506779.150) locus. Primers pMOtag forward and reverse were used to amplify the donor DNA from the pMOtag-23M plasmid [83], and the final PCR product was composed of 30 bp of the homologous arm (final 30 nucleotides from the H2B.V gene, excluding the stop codon), 3 copies of Myc epitope sequence, and the puromycin resistance gene and 30-bp homology arm (30 nucleotides from the intergenic region adjacent to the cleavage region). The donor DNA was inserted in the 3' region of the H2B.V gene to be expressed at the C-terminal portion of H2B.V protein. Primers "TcH2Bv 3´ UTR sgRNA" and "sg scaffold" were used to amplify sgRNA. PCR products were purified (QIAquick PCR Purification Kit Protocol) and transfected into epimastigotes expressing Cas9 (CL Brener strain). A total of 10^8 epimastigotes were resuspended in 350 µL of transfection buffer (90 mM sodium phosphate, 5 mM potassium chloride, 0.15 mM calcium chloride, 50 mM HEPES, pH 7.2) together with 50 µL of purified DNA (PCR products for donor DNA and sgRNA). The 400-µL final volume was placed in 0.2-cm cuvettes (Bio-Rad) and electroporated in the Bio-Rad Gene pulser with 2 pulses of 500 µF and 450 V. After electroporation, parasites were placed in fresh LIT medium with 10% SFB at 28°C. Then, after 24 h, puromycin was added to the medium at a final concentration of 10 µg/mL. After approximately two weeks, the polyclonal population was selected and then cloned. Epimastigotes were diluted to 1 parasite per mL in LIT/10% SFB containing puromycin and then plated in 96-well plates (200 µL per well). The parasites were incubated at 28°C with 5% $CO_2$, and monoclonal populations were obtained within approximately 3 weeks. Clones were obtained by serial dilution. Genome edition and allele-specific insertion were confirmed by PCR using the primers

Fw_mycH2BVcheck_Chr27P (1), Fw_mycH2BVcheck_Chr27S (2), Rv_mycH2BVcheck_myc (3), and Rv_mycH2BVcheck_puromicin (4) (S1 Table).

## Cloning and recombinant protein expression

All primer sequences are described in S1 Table. The H2B gene (TcCLB.511635.10), BDF2 (TcCLB.506553.20) and H2A.Z (TcCLB.511323.40) from *T. cruzi* (CL Brener strain) were amplified using the following pair of primers (all containing Eco RI and Hind III cleavage sites): H2Bc_F and H2Bc_R; HA_BDF2_Forward and BDF2_Reverse, and H2A.Z_Forward and H2A.Z_Reverse. BDF2 was amplified by fusion with HA from the vector pDEST17, which was kindly provided by Dr. Esteban Serra. The PCR products were purified and cloned first at pJET1/2blunt (Invitrogen), and after SANGER sequencing confirmation, the insert was sub-cloned either at pET28(a)+ (Novagen) (for H2B and BDF2) or at pMAL-p2x (NEB) (for H2A. Z). Plasmids were transfected into *E. coli* DH5α and/or at BL21 DE3 for recombinant expression as described previously [36]. Briefly, *E. coli* BL21 DE3 transformed either with pET28a (+)-H2B.V [36], pET28a(+)-H2B, pET28a(+)-BDF2 or pMAL-p2x-H2A.Z were inoculated in LB base medium (tryptone 1%, NaCl 1%, yeast extract 0.5%, pH 7) containing kanamycin 50 μg/mL at 37˚C for up to 18 h, and recombinant expression was induced with 1 mM IPTG. H2B, H2B.V and BDF2 (His-tag) and H2A.Z (fused with a maltose-binding protein -MBP) purification were performed by affinity chromatography using Ni-NTA agarose (Qiagen) and amylose resin (NEB), respectively, according to the manufacturer's recommendations.

## Cell cycle and EU (5-Ethynyl-2'-uridine) assay

Epimastigotes in the exponential growth phase were fixed in 70% ethanol overnight at -20˚C and treated with 10 μg/mL RNase A (Thermo Scientific) and 40 μg/mL propidium iodide (Sigma-Aldrich, St. Louis, MO). Cells were analyzed using the Attune® Acoustic Focusing Cytometer (Applied Biosystems). Epimastigotes in the exponential growth phase (~5x10$^6$ cells/mL) were incubated with EU 10 μg/mL (Life) and fixed in cold 50% ethanol for 20 min, as described in [84]. Cells were analyzed using the Attune® Acoustic Focusing Cytometer (Applied Biosystems). Data from 20,000 events were analyzed using FlowJo v. 10.7.1 software.

## Protein extracts

A total of $5 \times 10^8$ parasites were used to obtain basic protein extracts [33]. Extracts were quantified using the Pierce BCA Protein Assay Kit protocol (Thermo Scientific) and stored at -20˚C.

## Pulldown assays

The pulldown protocol was based on the principle of immobilization of recombinant protein using Ni-NTA agarose resin (Qiagen). The experiments were performed in biological triplicates and designed to contain 4 control samples (resin Ni-NTA with i. bacteria extract, ii. Recombinant H2B. V or BDF2, iii. epimastigote extracts, iv. TCT extracts) and 2 experimental extracts (Ni-NTA resin i. epimastigote extracts and recombinant H2B.V or BDF2; ii. TCT extracts and recombinant H2B.V or BDF2). Recombinant protein was first incubated with nickel resins for 30 min under slow stirring at 4˚C, followed by a 1-h incubation with parasite basic protein extracts (described above). The resin was washed three times with TBS, and recombinant protein and its interactors were eluted with 50 mM sodium phosphate, 0.3 M NaCl and 290 mM imidazole buffer. One-third of the eluate was reserved for fractionation by

SDS-PAGE, and the remainder was reserved for proteomics analysis. The BDF2 eluates were analyzed by western blotting.

## Proteomics analysis

Pulldown eluates and parasite extracts (basic protein extracts from H2B.V-HtzKO parasites) were precipitated with 20% TCA and resuspended in 8 M urea, 75 mM NaCl, and 50 mM Tris, pH 8.2. Protein was reduced with 5 mM DTT and alkylated with 14 mM iodoacetamide followed by digestion with trypsin (1:100, w:w) (Promega) as described in [85]. The peptides were resuspended in 0.1% formic acid and injected (5 μl) into a precolumn (Thermo—C18 of 5 μm, 2 cm x 100 μm) coupled to a nano HPLC (NanoLC-1DPlus, Proxeon). The separation was carried out in a capillary precolumn (10 cm x 75 μm containing C18 resins of 3 μm) in a 2–50% gradient of acetonitrile in 0.1% formic acid for 1 h. The eluted peptides were analyzed directly on an Orbitrap spectrometer (LTQ-Orbitrap Velos-Thermo). The 10 most intense ions were selected for fragmentation by CID. All ions were analyzed in positive ionization mode. The data in * raw format were processed using the Andromeda-MaxQuant program [86] with TriTrypDB (*T. cruzi* taxonomy—obtained from http://tritrypdb.org/tritrypdb/). Database searches were performed using the following parameters: carbamidomethylation of cysteine as a fixed modification; oxidation of methionine, N-terminal acetylation as variable modifications; tolerance MS1 of 6 ppm and MS2 of 0.5 Da, and 1% FDR. The ProteinGroups. txt output was sequentially processed using the Perseus program [86]. Proteins with LFQ intensity values equal to zero were considered absent. Proteins considered by MaxQuant as contaminants, presented in the reverse database and identified only by a modified peptide, were removed. For pulldown analysis, we also removed proteins presented (LFQ value > 0) in only one of the 3 biological replicates; proteins with a ratio of LFQ values from the experimental sample (EpiH2B, TripoH2B, EpiH2BV and TripoH2BV) to the control sample (Control_Epi, Control_TCT) lower than 1.5 (to ensure that eluted protein was enriched in the pulldown assay over the control samples); and common contaminant proteins (ribosomal, cytoskeleton, heat-shock and metabolic proteins). Finally, we only considered proteins that were present in a previous chromatin study [36] and those with *T. brucei* protein orthologs located in the nucleus/chromatin according to TrypTag. For histone quantitation on H2B. V-HtzKOs, the iBAQ values for all detected histones were considered 100%.

## ChIP-seq

ChIP-seq experiments were performed as described in [16] with a few modifications. Epimastigote and TCT H2B.V-myc-tagged cells (1x10$^8$) were cross-linked with 1% formaldehyde for 20 min at RT and quenched with 2 M glycine followed by washes with cold TDB buffer (5 mM KCl, 80 mM NaCl, 1 mM MgSO$_4$, 20 mM Na$_2$HPO$_4$, 2 mM NaH$_2$PO$_4$, and 20 mM glucose (pH 7.4)). The permeabilization buffer and NP-S buffer described in [16] were substituted for lysis buffer (1 mM potassium L-glutamate, 250 mM sucrose, 2.5 mM CaCl$_2$, 1 mM PMSF) [36]. Upon crosslinking, cells were resuspended in 1 mL lysis buffer, centrifuged, and resuspended in lysis buffer containing 0.1% Triton X-100 for 15 min at RT. Then, the cells were pelleted, washed, and incubated with 75 U of Mnase (Sigma Aldrich, #N3755) at 25˚C for 10 min. After the addition of 10 mM EDTA to block digestion, the supernatant was collected and washed with lysis buffer with 0.1% SDS. The pellet was sonicated in a Covaris S220 (1 min, 10% duty factor, 200 cycles per burst). Sheared DNA was centrifuged (10,000 x g for 10 min at 4˚C). Fifty microliters of Dynabeads protein G (Thermo Fisher) was resuspended in PBS-Tween (0.02%) containing 10 μg of purified anti-Myc antibody (Helmholtz Zentrum München, Monoclonal Antibody Core Facility) and incubated with slow rotation at 4˚C

overnight. Antibody-coupled beads were incubated at 4˚C overnight with slow rotation with the MNase supernatant obtained above, washed eight times with cold RIPA buffer (50 mM HEPES-KOH pH 7.5, 500 mM LiCl, 1 mM EDTA, 1% NP-40, 0.7% Na-deoxycholate), washed one time with TE buffer containing 50 mM NaCl and eluted with 50 mM Tris-HCl pH 8.0, 10 mM EDTA, 1% SDS at 65˚C for 30 min. The crosslinking of ChIP and input samples was reversed with 300 mM NaCl (at 65˚C for 9 h) and treated with RNase A and proteinase K as described. After DNA purification, ChIP and input DNA samples were used to construct an Illumina library using TrueSeq adapters according to [16]. The libraries were sent for Illumina NextSeq sequencing with 75-bp paired-end sequencing at the Core Unit Systems Medicine, University of Würzburg.

### ChIP-seq analysis

Quality filtering and adapter trimming of the Illumina sequencing reads were performed with fastp version 0.20.0 [60]. Reads with more than 5 N bases were discarded, as were those with more than 40% of their bases with Phred quality values below 15 ($< Q15$). All nucleotides from the 5' or 3' ends of each read with quality inferior to 5 ($< Q5$) were removed. Since these data were paired-end, we activated the correction algorithm: for size overlaps of at least 30 nucleotides (accepting 5 mismatches), a base is corrected if it is ultralow quality ($< Q15$), while the corresponding base at the mate read is high quality ($> Q30$). After the application of all these filters, reads shorter than 15 base pairs were removed. Approximately 94% of the reads passed the described filters and were used for downstream analysis. Quality control was performed with FastQC version 0.11.8 [61]. The local mapping of the filtered reads against the genome of the CL Brener Esmeraldo-like strain (release 32) and against the TCC strain (release 44) from TriTrypDB was performed with bowtie2 version 2.3.5.1 [62], with parameters more restrictive than the preset option: very-sensitive-local parameters (-D 25 -R 4 -N 1 -L 19 -i S,1,0.40—nceil L,0,0.15). Coverage plots were generated using COVERnant v.0.3.2 (https://github.com/konrad/COVERnant) and visualized using IGV (http://software.broadinstitute.org/software/igv/). COVERnant ratios were obtained by pairwise analysis of each ChIP and input sample. For the identification of genomic regions enriched in H2B.V, the bam files were analyzed using the findPeaks (-sytle histone), getDiffPeaks (-style histone) and mergePeaks (-d given [default] = maximum distance to merge) commands from HOMER [64] with default options. Peaks were obtained by the findPeaks command considering a fold (-f) enrichment of 2 or 4 over the corresponding input sample. To analyze histone enrichment at specific genomic regions, genomic coordinates of each feature were obtained from an in-house customized GFF file. Summary plots and heatmaps were obtained using the computeMatrix (with scale-regions and skipZeros options) and plotHeatmap (with hierarchical clustering) functions from deepTools2 [87,88]. The GFF file from *T. cruzi* (available at https://tritrypdb.org/) was customized in-house to contemplate the polycistrons, dSSRs and cSSRs.

### WB and immunofluorescence (IFA)

WB and IFA were performed as described with antibodies, and the dilutions used were as follows: anti-Myc (Cell Signaling) 1:3000; anti-HA (Cell Signaling) 1:3000, anti-H3 (Abcam) 1:10000, anti-BIP 1:1000 (polyclonal), anti-H2A. Z (polyclonal–kindly provided by Dr. Nicolai Siegel), anti-His Tag 1:3000 (Cell Signaling); and anti-GAPDH 1:4000 (polyclonal) [89]. For IFA of myc-tagged parasites, after fixation with 4% paraformaldehyde and permeabilization with Triton X-100, an overnight incubation with anti-Myc (Cell Signaling) antibody 1:3000 in 1% PBS/BSA at 4˚C followed by secondary antibody conjugated to Alexa Fluor 555 (Thermo Fisher) at 1:500 in PBS/BSA 1% was performed. When mentioned, IFA using anti-GP90 and

anti-GP82 1:1000 (kindly provided by Nobuko Yoshida) was performed as described previously [90]. Slides were further mounted with Vectashield plus DAPI (Vector laboratories) and visualized at 100 X magnification in an inverted OLYMPUS microscope model IX81 with Z axis motorization. Images were acquired using OLYMPUS CELL R version software 3.2.

## Quantitative PCR

TRIzol Reagent (Invitrogen) was used to isolate total RNA from *T. cruzi* epimastigotes (Y strain–wt and HtzKO cl2 and cl4; and CL Brener strain–wt, Cas9, H2B.V-myc tagged) in exponential growth phase. Next, 10 μg from each total RNA sample was subjected to DNAse I (Sigma) treatment to remove genomic DNA contamination. Then, 1 μg of purified RNA was used for cDNA synthesis with a Super Script Reverse Transcriptase Kit (Invitrogen) following the manufacturer's recommendations. Quantitative real-time PCRs were performed using H2B.V_qPCR_F and H2B.V_qPCR_R as the target genes, and GAPDH_Tcruzi_qPCR_Fw and GAPDH_Tcruzi_qPCR_Rv were used as endogenous control genes (S1 Table). For each reaction, 5 μL of 2X Power SYBR Green PCR Master Mix (Applied Biosystems), 2 ng cDNA from wt and HtzKOs (Y strain); and 10 ng cDNA from wt, Cas9 and H2B.V-myc tagged (CL Brener strain); and 300 nM H2B.V primer or 100 nM GAPDH primer were used. All reactions were carried out in triplicate in a 96-well plate. The plates were run on a StepOnePlus Real Time PCR System (Applied Biosystems). The relative quantification of H2B.V transcripts was calculated by the $2^{-\Delta Ct}$ method [91].

## Faire assay

The FAIRE samples were prepared as described in [92] with some modifications. The $2 \times 10^8$ epimastigotes of wild type, H2B.V HtzKO cl2 and cl4 were fixed with 37% formaldehyde and directly added to LIT medium to a final concentration of 1% for 5 minutes at RT. The formaldehyde was quenched with 2.5 M glycine to a final concentration of 125 mM. Then, the parasite pellets were resuspended in 2 mL of cold TELT buffer (50 mM Tris-HCl pH 8.0, 62.5 mM EDTA, 2.5 M LiCl, 4% Triton X-100) and sonicated in a Tommy Ultrasonic Disruptor UD-201 apparatus with an output of 4 and duty of 30 for 10 cycles of 30 seconds with 1-minute intervals. To prepare input control DNA, one-fourth (500 μL) of each sample was incubated with 5 μL of proteinase K for 1 h at 55°C, and the cross-linking was reversed by overnight incubation at 65°C. After that, the DNA was extracted with phenol:chloroform:isoamyl alcohol solution and precipitated with 3 M sodium acetate (pH 5.2) and 95% ethanol. The input control DNA obtained was quantified using a NanoDrop (Thermo Fisher). To verify the sonication efficiency, 500 ng of input DNA control was run on a 1% agarose gel. To prepare FAIRE DNA, half of the fixed cell lysate (1 mL) was used. DNA was extracted as described above; however, protein removal by proteinase K and cross-linking reversal were performed after DNA purification. Then, DNA was purified with a MinElute PCR Purification Kit (Qiagen) following the manufacturer's recommendations. To assess the DNA yield, the total quantity of control DNA or FAIRE DNA (in ng, obtained by Nanodrop quantitation) was divided by the volume of cell lysate used for control DNA or FAIRE DNA obtention (in μL), respectively. Considering the control DNA yield as 100%, the retrieval ratio of FAIRE DNA for each sample was calculated.

## Sequential salt extraction of chromatin proteins

$10^8$ epimastigotes and TCTs H2B.V-myc tagged (CL Brener strain) were collected and lysed with extraction buffer (0.1% Triton X-100, 10 mM Tris-HCl [pH 7.4], 100 mM NaCl, 300 mM sucrose, 3 mM MgCl$_2$, 50 mM NaF, 1 mM Na$_3$VO$_4$, 0.5 mM phenylmethylsulfonyl fluoride,

and EDTA-free Complete protease inhibitor cocktail [Roche]), described at [93] for 10 min at 4°C under agitation. The samples were pelleted, and the supernatants were saved as soluble fraction (soluble fraction 1 –SF1). This step was repeated (soluble fraction 2 –SF2). The pellet was then treated with a sequential salt extraction gradient of 0, 200, 400, 600 mM de NaCl in mRIPA buffer, and the supernatants collected as described at [37]. The extraction was performed in biological duplicates and submitted to Western blot. The protein extracts equivalent to 200 ng of DNA (measured by Qubit) for Epimastigotes and TCTs were fractionated by SDS-Page 12%.

## Supporting information

**S1 Fig. H2B.V-myc tagged parasites generated by CRISPR-Cas9.** A. Schematic representation of CRISPR/Cas9 gene editing of the H2B.V gene (TcChr27-S:430,316–430,747). The small black arrow indicates the cleavage site caused by the Cas9 enzyme. IGR represents the intergenic region, and HArm represents the homologous arm. Donor DNA was amplified by PCR from the pMOTag23M plasmid using long primers. Long primer sequences are composed of 30 nucleotides corresponding to the homologous arm (5' end of the primer) plus 20 nucleotides complementary to the plasmid (3' end of the primer). The resulting PCR product, called donor DNA, is composed of 30 bp of the homologous arm of the H2B.V gene (in blue); 3 copies of myc sequence (in green); the *T. brucei* tubulin intergenic region (TIGR—dark gray); the resistance gene to puromycin (in yellow); and 30 bp of the homologous arm (in blue) from the 3' intergenic region (just after the cleavage site). The final edited locus H2B.V–myc is illustrated at the bottom of the scheme.The red arrows indicate the primers used to evaluate genomic edition in both alleles (Esmeraldo-like and Non Esmeraldo-like haplotypes). B. Agarose 1% gel showing that insertion of myc-tagged occurred in both alleles of the CL Brener strain. An insert of ~800 bp (primers 1+4 and 2+4), and 350–400 (1+3 and 2+3) were detected. Primers 1 and 2 were designed to be allele-specific. Genomic DNA from wt and Cas9 parasites were used as a negative control for amplification of myc and puromycin gene (C-) and as a positive control (C+) for amplification of H2B gene (primers H2Bc_F and H2Bc_R). C. Western blot assays confirm the expression of H2B.V-myc in epimastigotes maintained with puromycin (10–30 μg/mL). Untransfected parasites were used as a control (Ctl). Phenotypic evaluation of H2B.V-myc parasites: D. Growth curves (in log10) for Cas9 and H2B.V-myc parasites; E. The number of metacyclic trypomastigotes from Cas9 and H2B.V-myc parasites were counted in the RPMI supernatant after 8 days. The values were normalized to their respective numbers in wild-type cells. Experiments were performed in biological triplicates. No significant statistical differences. F. H2B.V relative expression levels were evaluated by qPCR in wild-type, Cas9 and H2B.V-myc epimastigote and H2B.V-myc TCTs. H2B.V transcript levels were assessed relative to GAPDH levels. No significant statistical difference. G. IFA showing the presence of H2B.V-myc in epimastigote and TCT life forms (white arrows).
(TIF)

**S2 Fig. Mapping coverage and peaks in H2B.V ChIP-seq in CL Brener and TCC assemblies.** A. Total reads and overall alignment of input and H2B.V- ChIP samples in the CL Brener Esmeraldo-like haplotype (release 32) and TCC (release 44) assemblies. B. H2B.V peaks identified in epimastigotes and TCTs using a peak calling algorithm available in HOMER [27] considering a fold enrichment of 4 (default) and a required Poisson p-value over input = 1.00e-04. C. Comparison of peak (fold 4) width (in bp) among life forms and genome mapping. (D) Venn diagrams (available online at https://www.meta-chart.com/) comparing common and different peaks at fold 4 between life forms.
(TIF)

**S3 Fig. Representation of dSSR and cSSR on the TCC strain and signals at CDS polycistrons.** A. IGV snapshot of H2B.V and H3 enrichment at contig PRFC000010. Note that H2B. V, but not H3, is enriched at dSSR (rectangles with interrupted lines). Blue bar arrows indicate the transcription direction in each polycistron. Genes in the same polycistron are stained with the same color. Red and green bars represent, respectively, H2B.V-peaks (fold 4) obtained by HOMER in epimastigote and TCT life forms. Purple and yellow bars represent dSSRs and cSSRs, respectively. B. Heatmap plots and k-mean clustering of H2B.V and H3 ChIP-seq signals at CDS polycistrons (deeptools—scale region function) and their 10-kb upstream and downstream regions, mapped to the TCC assembly. Black regions represent polycistrons that are located at the border of the contigs and therefore have either no upstream or downstream regions. C. Distribution of H2B.V peaks (fold 4) in the 20 longer contigs (PRFC000001 to PRFC000020) of the TCC assembly.
(TIF)

**S4 Fig. Distribution of enrichment H2B.V peaks in dSSR and multigenic family members.** A. IGV snapshot of H2B.V and H3 reads in TcChr23-S showing an enrichment of H2B.V between the conserved (green) and disrupted (orange) genome compartments (rectangle with interrupted lines) and two distinct patterns at dSSRs (black and red arrows). Blue bar arrows indicate the transcription direction in each polycistron. Genes in the same polycistron are stained with the same color. Red and green bars represent, respectively, H2B.V -peaks (fold 4) obtained by HOMER in epimastigote and TCT life forms. Purple and yellow bars represent dSSRs and cSSRs, respectively. Black and red arrows indicate dSSRs with or without H2B.V enrichment, respectively. B. Heatmap plots and k-means clustering of H2B.V and H3 ChIP-seq signals in multigenic family members (deeptools—scale region function) and their 2-kb upstream and downstream regions mapped in the CL BrenerCL Brener- Esmeraldo-like and TCC assembly. C. Distribution of multigenic family members in clusters 1 to 3 described in B (dark blue, light blue and yellow lines in the summary plot). Note that the distribution of these genes in clusters 1 and 2 differs from the expected genome distribution (last bar).
(TIF)

**S5 Fig. Examples of enriched H2B.V signal in tDNA, rDNA clusters and dSSR.** IGV snapshot of H2B.V enrichment at tDNA loci, dSSR (A) and at rDNA (B) in the CL Brener Esmeraldo-like assembly. The enrichment at the indicated feature is highlighted by a rectangle with interrupted lines. Blue bars arrows indicate the transcription direction in each polycistron. Genes in the same polycistron are stained with the same color. Red and green bars represent, respectively, H2B.V -peaks (fold 4) obtained by HOMER in epimastigote and TCT life forms. Black bars indicate tDNA or rDNA.
(TIF)

**S6 Fig. H2B.V is more abundant but has lower binding affinity to chromatin in TCT forms.** WCE of TCTs and epimastigotes ($2x10^6$ parasites in quadruplicates) were probed against GAPDH (housekeeping gene), myc (H2B.V) and histone H3. Right, bands were quantified using ImageJ and plotted in Prisma. ** t-test (p-value < 0,005). B. Chromatin sequential salt extraction from epimastigotes and TCTs in biological duplicates were probed against BiP (cytosol marker), myc (H2B.V) and histone H3. SF1 –soluble fraction 1 and SF2 –soluble fraction 2 and pellet as insoluble sediment. The equivalent of 200 ng of DNA for epimastigotes and TCTs were fractionated in SDS-Page.
(TIF)

**S7 Fig. H2A.Z interacts with H2B.V *in vitro*.** Recombinant H2A.Z-MBP and recombinant H2B.V-His were incubated together or separately with amylose resin, known to specifically

interact with MBP. Eluates were fractionated in a 15% SDS-PAGE and transferred to nitrocellulose membranes. Western blot assay was performed using polyclonal antibodies anti-H2A.Z and anti-His (for H2B.V-HisTag). H2A.Z-MBP is ~60 kDa, and H2B.V-His is ~ 16.5 kDa. (TIF)

**S8 Fig. H2B.V-HtzKO construction and validation.** (A) Diagram of NEO cassette recombination at the H2B.V locus (Chr 27-S) in the *T. cruzi* CL Brener genome highlighting the location of H2B_EXT_FOR and H2B_EXT_REV primers used for amplification of the whole NEO cassette, which includes the 5' and 3' fragments of H2B.V. Lower, the location of primers H2B.V EXT_FOR (forward); Neo F (forward); Neo Out R (reverse); Neo Out F (forward); Neo R (reverse); and H2B.V EXT_REV (reverse) are highlighted. (B) pTc2KO-H2B.V-neo cassette insertion confirmation. Left, the 1% TAE agarose gel showing the 3045-bp amplicon from the complete NEO cassette recombined at the H2B.V locus (primers H2B_EXT_FOR and H2B_EXT_REV). Right, pTc2KO-H2B.V-neo cassette insertion confirmation in clone 4. A 1% TAE agarose gel of amplificons obtained from PCR using the following pair of primers: 1- H2B.V EXT_FOR / Neo Out R (1052 bp); 2- Neo Out F / H2B.V EXT_REV (1376 bp); 3- H2B.V EXT_FOR / Neo R (1819 bp); 4- Neo F / H2B.V EXT_REV (1966 bp); H2B.V_KpnI_forward / H2B.V_XbaI_reverse (1384 bp) for confirmation of heterozygous knockouts. M(pb)-molecular weight marker in bp (1 Kb Plus Ladder–Invitrogen). (C) Relative abundance of histones in H2B.V-HztKO and wild-type parasites. Total basic extracts from H2B.V-HtzKO and wild-type parasites were evaluated by label-free quantitative proteomics. iBAQ values were obtained from each sample (in biological triplicates), and the relative values were considered. Unpaired t-test ($^{*}$ p-value $<$ 0.05). (TIF)

**S9 Fig. H2B.V-HtzKO phenotypic analysis.** A. Phase contrast images of H2B.V-HtzKO, and wild-type parasites. B. Biological replicates of growth curves (in $\log_{10}$) for wild-type and H2B.V-HtzKO (cl 2 and 4) parasites. C. Percentage of cell cycle phases in H2B.V-HtzKO clones and wild-type parasites. Bars represent the average from biological triplicates. Error bars represent SEM values. D. Histograms of EU-positive cells of H2B.V-HtzKOs (cl2 and 4), wild-type (orange) and unlabeled parasites (green). The mean and standard variation of the percentage of positive cells were plotted (triplicate values). One-way ANOVA of variance–unpaired-test. E. Quantitation of the open chromatin of H2B.V-HtzKOs and wild-type parasites obtained after the FAIRE protocol. The retrieval ratio refers to the percentage of open chromatin related to the total DNA content obtained for each sample. F. Percentage of parasites (epimastigote, intermediate and metacyclic trypomastigotes) in stationary phase culture of H2B.V-HtzKOs and wild-type parasites. Parasites were classified based on the position and morphology of the nucleus and kinetoplast as proposed previously [25]. The metacyclics markers GP90 and GP82 were also used to allow discrimination of life forms. (TIF)

**S1 Table. Sequence of primers.** (XLSX)

**S2 Table. H2B.V—enriched peaks (fold$>$4) at CL Brener Esmeraldo -like haplotype obtained by HOMER.** (XLSX)

**S3 Table. H2B.V—enriched peaks (fold$>$4) at TCC assembly obtained by HOMER.** (XLSX)

**S4 Table. Pulldown assay—quantitative data (LFQ values) from all samples.**
(XLSX)

**S5 Table. Final list of H2B.V and H2B interactors.**
(XLSX)

## Acknowledgments

We thank Ivan Novaski Avino and Ismael Feitosa Lima for technical assistance and Alex Ranieri Lima for important inputs on the data and bioinformatic analysis. We thank Esteban Serra for the BDF2 plasmid, Roberto Docampo for the pMOtag23M plasmid, and Nobuko Yoshida for antibodies against metacyclic markers (GP90 and GP82). We thank Herbert Guimarães de Sousa Silva and Ana Paula de Jesus Menezes for reading this manuscript and providing critical comments.

## Author Contributions

**Conceptualization:** Simone Guedes Calderano, Maria Carolina Elias, Julia Pinheiro Chagas da Cunha.

**Data curation:** Juliana Nunes Rosón, Marcela de Oliveira Vitarelli, David da Silva Pires, T. Nicolai Siegel, Maria Carolina Elias, Julia Pinheiro Chagas da Cunha.

**Formal analysis:** Juliana Nunes Rosón, Marcela de Oliveira Vitarelli, Héllida Marina Costa-Silva, Kamille Schmitt Pereira, Leticia de Sousa Lopes, Barbara Cordeiro, Amelie J. Kraus, Simone Guedes Calderano, Stenio Perdigão Fragoso, Julia Pinheiro Chagas da Cunha.

**Funding acquisition:** Julia Pinheiro Chagas da Cunha.

**Investigation:** Juliana Nunes Rosón, Héllida Marina Costa-Silva, Kamille Schmitt Pereira, David da Silva Pires, Leticia de Sousa Lopes, Barbara Cordeiro, Amelie J. Kraus, Karin Navarro Tozzi Cruz, Simone Guedes Calderano, Stenio Perdigão Fragoso, T. Nicolai Siegel.

**Methodology:** Juliana Nunes Rosón, Stenio Perdigão Fragoso, T. Nicolai Siegel.

**Resources:** Julia Pinheiro Chagas da Cunha.

**Software:** David da Silva Pires.

**Supervision:** Simone Guedes Calderano, Stenio Perdigão Fragoso, T. Nicolai Siegel, Maria Carolina Elias, Julia Pinheiro Chagas da Cunha.

**Writing – original draft:** Juliana Nunes Rosón, Maria Carolina Elias, Julia Pinheiro Chagas da Cunha.

**Writing – review & editing:** Juliana Nunes Rosón, Marcela de Oliveira Vitarelli, Héllida Marina Costa-Silva, T. Nicolai Siegel, Maria Carolina Elias, Julia Pinheiro Chagas da Cunha.

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
