## [Decision Letter · Decision Letter 0]

17 Aug 2021

Dear Dr da Cunha,

Thank you very much for submitting your manuscript "­­ Histone H2B.V demarcates strategic regions in the Trypanosoma cruzi genome, associates with a bromodomain factor and affects parasite differentiation and host cell invasion" for consideration at PLOS Pathogens. As with all papers reviewed by the journal, your manuscript was reviewed by members of the editorial board and by several independent reviewers. In light of the reviews (below this email), we would like to invite the resubmission of a significantly-revised version that takes into account the reviewers' comments.

We cannot make any decision about publication until we have seen the revised manuscript and your response to the reviewers' comments. Your revised manuscript is also likely to be sent to reviewers for further evaluation.

Sincerely,

Lyris Martins Franco de Godoy, Ph.D.

Guest Editor

PLOS Pathogens

David Sacks

Section Editor

PLOS Pathogens

Kasturi Haldar

Editor-in-Chief

PLOS Pathogens

orcid.org/0000-0001-5065-158X

Michael Malim

Editor-in-Chief

PLOS Pathogens

orcid.org/0000-0002-7699-2064

Reviewer's Responses to Questions

**Part I - Summary**

Reviewer #1: The manuscript presents a novel and significant study of H2B.V histone variant in T. cruzi, assessing its genome distribution by CHIP-seq of endogenously tagged (Myc) parasites, its in vitro protein partners by affinity chromatography and its functional significance by in vitro assays of hemi-KO parasites.

The genome-wide distribution of histone variant H2B.V in T. cruzi epimastigotes uncovered a specific positioning at the dSSR of polycistron (as shown in T. brucei), the boundaries between conserved and disrupted gene compartments, and tDNA putative internal polycistronic TSS mostly. Curiously, cellular trypomastigotes are depleted of the variant in those regions and no specific enrichment could be found in this stage. Finally, H2B.V in vitro protein partners identified include BDF2 and H2A.Z (previously known to overlap in the SSR of T. brucei), and 10 H2B.V specific chromatin associated proteins.

Overall, these findings confirm the T. brucei model of H2B.V function, in which pol II transcription start sites (TSSs) are occupied by unstable nucleosomes containing H2A.Z and H2BV (as well as BDF3 and H4K10ac), likely facilitating transcription initiation.

The authors also study the effect of the single copy KO of the H2B.V variant, unable to analyze the double KO for the probable gene essentiality, not surprisingly since it H2B.V and H2A.Z were already shown to be essential for cell viability in T. brucei (2005) and L. major (2013). Interestingly, they see an alteration of the differentiation and invasion rates but not proliferation, growth, or transcriptional levels in the H2B.V-KO parasites.

Finally, the current manuscript provides novel chromatin information for T. cruzi gene families relevant for host infection, showing that gene in the MUCINs and MASP (located in the disrupted compartments are more enriched in H2B.V compared to the multigenic families TS, GP63, RHS, and DGF-1, perhaps for they preferential location at the first CDS of their cognate polycistron (as discussed by the authors).

The experimental strategy, experimental design, presentation and analysis of the data are correct, and the results are sound, and reasonable for the aim of this kind of high throughput studies. Future experiments would be needed to validate the findings and investigate the influence of H2B.V in gene transcription and its molecular mechanism of action. Additionally, the text is clear, the results are correctly presented, and the discussion is fair and honestly states the limitations of the study.

Reviewer #2: This is a comprehensive contribution to the understanding of chromatin status and dynamics in Trypanosoma cruzi that may represent an important step on gene expression regulation across parasite differentiation. The authors show that the distribution of the H2B-V demarcates strategic regions of the genome of T. cruzi, which affects replicative to nonreplicative differentiation and host cell invasion. Data presented is robust and thoroughly discussed. It may be considered a major contribution to the field.

**Part II – Major Issues: Key Experiments Required for Acceptance**

Reviewer #1: A major concern is the emphasis made by the authors narrative on the replicative vs non-replicative states. Since replication is not the major difference between epimastigotes and trypomastigotes forms, and the functional results of this study point to stage specific roles of H2B.V and not proliferative ones, it is unclear why the authors highlight the replicative comparison. The authors need to clarify the relevance of the H2B.V replicative connection and incorporate amastigotes and metacyclic trypomastigote in the study. If that stress is just a narrative imprecision, the authors should rephrase the associated paragraphs.

The second major concern is about the expression of H2B.V in trypomastigotes. In De Jesus et al 2016, the authors found that H2B.V variant is notoriously overexpressed in trypomastigotes compared to epimastigotes, as mentioned in the manuscript. In the current study they show the same trend trough the quantification of myc-H2B.V expression by western blot in whole extracts. Yet, SF1C shows myc-H2B.V trypomastigote parasites (probably indicated by the arrows -although there is no reference to them in the legend or the main text-), lacking or having faint myc signal. Unfortunately, Figure SF1C is not commented in the text. Although the authors extensively discuss the conundrum of the high expression of H2B.V but no enrichment in chromatin regions (364-376), the SF1C pictures (if correct) are impressive. The authors must clarify how is the expression myc-H2B.V in the edited trypomastigote parasites using IHQ. In addition, an assessment of the expression of myc-H2B.V in the transfected cell line compared to the wild type by qRT-PCR is necessary. Also, it is unclear if the edited parasite line is homozygote for the tag, and that can be also discriminated by qRT-PCR. Although the authors explain the lack of antibodies (138 lines), allele specific PCR is possible with this type of edition, and qRT-PCR just to assess the bi-allelic expression can be done as for the KO transfectants.

Reviewer #2: (No Response)

**Part III – Minor Issues: Editorial and Data Presentation Modifications**

Reviewer #1: I suggest changing the title entirely. The adjective “strategic” has an overly broad meaning and does not effectively communicate the findings of the study. In addition, the emphasis in the interaction with the bromodomain protein points to a finding that is not central in the manuscript and may need further proof to become meaningful in vivo and for the 2 stages analyzed.

Line 41- …was preferentially located at divergent “TRANSCRIPCIONAL STRAND” switch regions. Insert the words in upper case or similar to be precise.

Although it is common to omit this aspect, I think is always useful to justify the selection of parasite strains for the experiments. Please comment the reason of the choice of the 2 different strains used. Likewise, justify the analysis of TCC instead of Dm28d of the 3rd generation T. cruzi genomes.

The word “polycistron-CDS” in unusual. May the authors define with accuracy what they mean by it?

Figure 1. The legend of B and C is confusing. The DNA window of the summary plot is unclear (2kb?, 10kb?, distance start-end?). Please state the size of the region presented in each case.

Samples E1 and T1 are similar in the summary plots. Could the authors provide an explanation for this? If the authors have a quantification of the parasite forms in the parasite culture obtained in each sample, it would be worth to include it.

An analysis of the H2B.V parasites in terms of proliferation and differentiation in comparison to the wild type strain would be useful to evaluate putative alterations of the edited transfectants.

Figure S2 B- Amend the color indications.

Line 242-3. Although the authors state “H2B.V interacts with more proteins in TCT”, this is also true for H2B; then it might be the H2B interactome and not the differential role of H2B.V in epimastigotes and TCT the source of variation. Please rephrase to avoid the misleading meaning.

Line 277. It is unclear what pull down (our) are the authors referring to in the sentence.

Having a myc-H2B.V edited strain, one wonder why the authors did not perform the identification or the validation of protein partners in vivo using the myc tag.

Figure 3B controls are different for E and TCT. Please clarify the reason.

The experiment shown in Figure S9 could be in the main article since it is directly related to the hypothesis, and I think would yield statistically significant differences if more replicates are incorporated.

Reviewer #2: I will raise some issues which I think should be modified or better addressed in the manuscript:

I would suggest modifying the title to: Histone H2B.V demarcates strategic regions in the Trypanosoma cruzi genome and affects parasite differentiation and host cell invasion. Although interesting, the interaction of H2B.V with a bromodomain factor, it was not functionally explored and several other potentially interesting interactions were equally witnessed, so I see no point in including this information in the title.

The authors state that “To improve the data analysis and biological conclusions, the reads were mapped against two T. cruzi genome assemblies (the Esmeraldo-like haplotype of the CL Brener strain and against the TCC 148 strain)…”. The comparative analysis was thoroughly described in results, but poorly discussed and a wrap up on the findings and relevance of the comparison is missing in the Discussion.

The counterintuitive results, quite interesting and intriguing, on the amount of H2B.V protein versus the profile of deposition on the genome in epimastigotes and TCT should be better explored. The authors focus the discussion on the chromatin structure and histone binding features, association of H2B.V with H2A.Z (as observed in T. brucei), or differences on PTMs (starting on line 367). Possible moonlight activities are mentioned later on in the discussion (line 434), and it seems to me that an unknown function played by this histone variant could be a possible explanation for the abovementioned results. Thus, the connections of their findings and possible moonlight functions should be better explored in the discussion. Moreover, it is necessary to refer to at least one of the several articles presenting the known “moonlight proteins” (or functions) and if histones have ever been shown to harbor moonlight functions.

The discussion on the differences of active chromatin levels in the stages studied are not supported by the presented data, it requires further experiments/analysis.

Finally, the authors forgot to mention that post-transcriptional mechanisms are considered to be the center of gene expression regulation in these parasites. It seems proper to refer the corresponding literature; it would highlight the relevance of their own studies.

Minor points:

Line 100, page 4 – Leishmania is not italicized.

Results:

Line 162, page 6 – by 2SC would the authors mean 2C?

Lines 164-165. Authors should choose between better explaining the cSSR and tDNA information at this point of the manuscript (including figure panel) or not mentioning it here to leave the information for when presenting the tDNA itself.

Line 168, page 6 – there is an s to be removed from S2B-D Fig

Line 197 – in the sentence “.. preferentially located at the first CDS”, wouldn’t it be good to add “of a given PTU”.

Lines 271-276 – The conclusion on this paragraph is a too strong statement considering that they have only evaluated histone H3.

Lines317-319 – S9E panel is not presented. Apparently, the S9E panel is presented as S9D, and if the mentioned “decreasing trend” the authors mention refers to panel D, any conclusion would need further experiments/analysis as mentioned in the paragraph that follows (starting on line 320). Thus, I would suggest to either include the mentioned analysis or to present this result as inconclusive, avoiding the bias of mentioning a “decreasing trend” that needs further analysis for a firm conclusion.

S5 Figure is not cited anywhere in Results, only en passant in the Discussion. I suggest inserting the information it contains in the Results section.

S8 Fig. – Panel A description and/or representation must be improved. The colors used (blue, gray, pink and orange) in the lower scheme do not help to understand where they came from.

Discussion:

Lines 368-370 – The sentence should be rephrased.

Line 416 – before the i. there is a punctuation missing.

Line 417 – fulfill misses one l

Line 417/418 – The reduced level should go with was

References:

21 among others are bugged – with unrecognized characters

Legend figures –

As a general comment for the legends: in the IGV outputs, frequently, if not always, the authors included a rectangle with interrupted lines. Although the meaning may be obvious, the authors must explain in the legend what these boxes are demarcating.

Fig 1- Improving legend is necessary:

Start-End notation probably corresponds to the S and E indicated at the graph on panels B and C. Does it mean in the case of CDS (B), Start and End of a CDS? And would it indicate start and end of an dSSR or cSSR in panel C?

Text is truncated at line 986.

Fig 2 – There should be added a “respectively” to the sentence on lines 992-994

S2 Fig.

Line 1049 – Brener is misspelled.

It seems that this legend does not correspond to the S2 figure presented.

S4 Fig

Text included at the bottom of panel B is truncated.

PLOS authors have the option to publish the peer review history of their article (what does this mean?). If published, this will include your full peer review and any attached files.

Reviewer #1: No

Reviewer #2: No
---

## [Editor Report · Decision Letter 1]

31 Jan 2022

Dear Dr da Cunha,

We are pleased to inform you that your manuscript 'H2B.V demarcates divergent strand-switch regions, some tDNA loci and genome compartments in Trypanosoma cruzi and affects parasite differentiation and host cell invasion' has been provisionally accepted for publication in PLOS Pathogens.

Best regards,

Lyris Martins Franco de Godoy, Ph.D.

Guest Editor

PLOS Pathogens

David Sacks

Section Editor

PLOS Pathogens

Kasturi Haldar

Editor-in-Chief

PLOS Pathogens

orcid.org/0000-0001-5065-158X

Michael Malim

Editor-in-Chief

PLOS Pathogens

orcid.org/0000-0002-7699-2064
---

## [Editor Report · Acceptance letter]

15 Feb 2022

Dear Dr da Cunha,

We are delighted to inform you that your manuscript, " H2B.V demarcates divergent strand-switch regions, some tDNA loci, and genome compartments in Trypanosoma cruzi and affects parasite differentiation and host cell invasion," has been formally accepted for publication in PLOS Pathogens.

Best regards,

Kasturi Haldar

Editor-in-Chief

PLOS Pathogens

orcid.org/0000-0001-5065-158X

Michael Malim

Editor-in-Chief

PLOS Pathogens

orcid.org/0000-0002-7699-2064